# Active poly-GA vaccination prevents microglia activation and motor deficits in a *C9orf72* mouse model

Qihui Zhou[1,2,†] (iD), Nikola Mareljic[1,†], Meike Michaelsen[1], Samira Parhizkar[3], Steffanie Heindl[4], Brigitte Nuscher[3], Daniel Farny[1], Mareike Czuppa[1], Carina Schludi[1], Alexander Graf[5], Stefan Krebs[5], Helmut Blum[5], Regina Feederle[1,2,6] (iD), Stefan Roth[2,4], Christian Haass[1,2,3] (iD), Thomas Arzberger[1,2,7,8], Arthur Liesz[2,4] (iD) & Dieter Edbauer[1,2,9,*] (iD)

## Abstract

The *C9orf72* repeat expansion is the most common genetic cause of amyotrophic lateral sclerosis (ALS) and/or frontotemporal dementia (FTD). Non-canonical translation of the expanded repeat results in abundant poly-GA inclusion pathology throughout the CNS. $(GA)_{149}$-CFP expression in mice triggers motor deficits and neuroinflammation. Since poly-GA is transmitted between cells, we investigated the therapeutic potential of anti-GA antibodies by vaccinating $(GA)_{149}$-CFP mice. To overcome poor immunogenicity, we compared the antibody response of multivalent ovalbumin-$(GA)_{10}$ conjugates and pre-aggregated carrier-free $(GA)_{15}$. Only ovalbumin-$(GA)_{10}$ immunization induced a strong anti-GA response. The resulting antisera detected poly-GA aggregates in cell culture and patient tissue. Ovalbumin-$(GA)_{10}$ immunization largely rescued the motor function in $(GA)_{149}$-CFP transgenic mice and reduced poly-GA inclusions. Transcriptome analysis showed less neuroinflammation in ovalbumin-$(GA)_{10}$-immunized poly-GA mice, which was corroborated by semiquantitative and morphological analysis of microglia/macrophages. Moreover, cytoplasmic TDP-43 mislocalization and levels of the neurofilament light chain in the CSF were reduced, suggesting neuroaxonal damage is reduced. Our data suggest that immunotherapy may be a viable primary prevention strategy for ALS/FTD in *C9orf72* mutation carriers.

**Keywords** amyotrophic lateral sclerosis; *C9orf72*; frontotemporal dementia; immunotherapy; neurodegeneration
**Subject Categories** Immunology; Neuroscience

## Introduction

Amyotrophic lateral sclerosis (ALS) and frontotemporal dementia (FTD) cases with *C9orf72* mutation show characteristic nuclear foci of the sense and antisense repeat RNA ($(G_4C_2)_n$ and $(C_4G_2)_n$) as well as neuronal inclusions resulting from non-canonical translation of the repeat RNA in all reading frames (DeJesus-Hernandez *et al*, 2011; Renton *et al*, 2011; Ash *et al*, 2013; Mori *et al*, 2013; Edbauer & Haass, 2016). In addition, *C9orf72* protein levels are reduced due to impaired transcription of the expanded allele (Frick *et al*, 2018). Gain-of-function mechanisms are likely the main drivers of *C9orf72* disease because *C9orf72* knockout mice develop mainly alterations of the peripheral immune system, reminiscent of systemic lupus erythematosus (Atanasio *et al*, 2016; Jiang *et al*, 2016). None of the *C9orf72*-specific pathologies correlate robustly with TDP-43 pathology or neurodegeneration suggesting synergistic and/or non-cell autonomous effects are crucial (Mackenzie *et al*, 2013, 2015; Schludi *et al*, 2015; DeJesus-Hernandez *et al*, 2017). So far, toxicity due to expression of individual dipeptide repeat (DPR) proteins has been studied most extensively. Poly-GA is the most abundant DPR species in patients and poly-GA expression has the strongest effect on TDP-43 phosphorylation and aggregation in primary neurons and mice, which correlates best with neurodegeneration in patients (Khosravi *et al*, 2017; Lee *et al*, 2017; Schludi *et al*,

1   German Center for Neurodegenerative Diseases (DZNE), Munich, Munich, Germany
2   Munich Cluster of Systems Neurology (SyNergy), Munich, Germany
3   Chair of Metabolic Biochemistry, Biomedical Center (BMC), Faculty of Medicine, Ludwig-Maximilians-Universität Munich, Munich, Germany
4   Institute for Stroke and Dementia Research, Ludwig-Maximilians-University Munich, Munich, Germany
5   Laboratory for Functional Genome Analysis, Gene Center, Ludwig Maximilian University of Munich, Munich, Germany
6   Monoclonal Antibody Core Facility and Research Group, Institute for Diabetes and Obesity, Helmholtz Zentrum München, German Research Center for Environmental Health (GmbH), Munich, Germany
7   Center for Neuropathology and Prion Research, Ludwig-Maximilians-University Munich, Munich, Germany
8   Department of Psychiatry and Psychotherapy, University Hospital, Ludwig-Maximilians-University Munich, Munich, Germany
9   Ludwig-Maximilians-University Munich, Munich, Germany
    *Corresponding author. Tel: +49 89 440046510; E-mail: dieter.edbauer@dzne.de
    †These authors contributed equally to this work

2017; Nonaka *et al*, 2018). In addition, poly-GA mouse models show motor deficits and neuroinflammation (Zhang *et al*, 2016; Schludi *et al*, 2017). Large biomarker studies and neuropathological reports from rare presymptomatic *C9orf72* cases suggest that all pathognomonic features of the disease are present decades before the onset of symptoms in the absence of widespread TDP-43 aggregation and neuron loss (Vatsavayai *et al*, 2016; Gendron *et al*, 2017; Lehmer *et al*, 2017). These prodromal changes likely initiate a slow disease cascade with neuroinflammation and TDP-43 aggregation that may eventually become independent of the initial trigger (Edbauer & Haass, 2016). Thus, targeting the *C9orf72*-specific pathomechanisms early will be essential for effective disease prevention or therapy.

Targeting cytoplasmic Tau aggregates by active and passive immunotherapy has shown promise in rodent models of Alzheimer's disease (AD), and effects on both intracellular and (unconventionally secreted) extracellular Tau have been discussed (Congdon & Sigurdsson, 2018). For example, intraventricular infusion of Tau antibodies reduced the aggregate burden and improved cognitive deficits in mice (Yanamandra *et al*, 2013). Moreover, active immunization targeting the Tau oligomerization domain was beneficial in a rat model and is being investigated in a phase 2 clinical trial (Kontsekova *et al*, 2014; Novak *et al*, 2017, 2018a; Jadhav *et al*, 2019). Unfortunately, the more advanced clinical trials with active and passive immunotherapy targeting extracellular Aβ aggregates in patients with AD were largely disappointing despite great promise from mouse models (Schenk *et al*, 1999; Sevigny *et al*, 2016). The key insights from these studies are as follows: (i) Antibodies can penetrate the blood–brain barrier at low levels that may be sufficient to affect the target (Schenk *et al*, 1999; Orgogozo *et al*, 2003; Landen *et al*, 2017). (ii) Clearance of Aβ load correlates with the antibody titer (Sevigny *et al*, 2016). (iii) Unwanted T-cell responses to active vaccination caused severe meningoencephalitis in a subset of AD patients but may be prevented by careful choice of the epitope (Schenk *et al*, 1999; Orgogozo *et al*, 2003; Axelsen *et al*, 2011). (iv) Limited cognitive improvement despite efficient clearance of Aβ in patients suggests disease mechanisms change and therapy should be started very early, maybe even in the prodromal phase, which is difficult in a mainly sporadic disease (Landen *et al*, 2017).

The common *C9orf72* mutation offers a unique opportunity for preventive therapy in ALS/FTD, but immunotherapy has not been tested *in vivo*. We and others have shown that cytoplasmic DPR proteins are transmitted between cells (Westergard *et al*, 2016; Zhou *et al*, 2017; Nonaka *et al*, 2018). After showing that monoclonal antibodies inhibit poly-GA aggregation and seeding from patient brain extracts *in vitro* (Zhou *et al*, 2017), we asked whether DPR immunotherapy would ameliorate DPR pathology and related deficits in an established poly-GA mouse model (Schludi *et al*, 2017). To this end, we tested different peptide formulations and analyzed the effects on motor deficits, poly-GA, TDP-43, neuronal damage, and neuroinflammation in vaccinated mice.

# Results

### Ova-$(GA)_{10}$ elicits strong antibody response in mice

To induce anti-GA antibody production *in vivo*, we immunized wild-type and GA-CFP mice prior to the onset of motor deficits with poly-GA peptides in incomplete Freund's adjuvant. Since poly-GA is expected to be poorly immunogenic, we used ovalbumin as an immunogenic carrier molecule (Ova-$(GA)_{10}$) that also may keep the aggregation-prone poly-GA soluble. In addition, we used a carrier-free self-aggregating $(GA)_{15}$ as immunogen (Chang *et al*, 2016). We started the immunization at 8 weeks of age and continued with boosting in 4-week intervals (Fig 1A) while subjecting the mice to weekly behavioral tests (see Fig 2). Immunization did not increase lethality or cause other obvious negative side effects in wild-type or transgenic mice (Fig EV1). Regardless of the immunization regimen, the mice gained weight steadily (Fig EV1A). To quantify the humoral immune response, we collected serum from all mice 1 week before initial immunization and 1 week after each immunization. We quantified the anti-GA IgG immune response in the mouse sera using an ELISA with GST-$(GA)_{15}$ as antigen and a mouse monoclonal anti-GA antibody as reference standard. Surprisingly, $(GA)_{15}$ induced no detectable antibody response as compared to the control mice that received PBS in adjuvant even after multiple boosting. In contrast, Ova-$(GA)_{10}$ induced a strong antibody response increasing up to ~ 400 μg/ml after the fourth boosting in both wild-type and GA-CFP transgenic mice (Fig 1B). Despite the strong immune response, distribution of splenic lymphocyte population was not affected in immunized mice (Fig EV1B).

The Ova-$(GA)_{10}$ antiserum specifically detected poly-GA in transfected HEK293 cells by immunoblotting and immunofluorescence (Fig EV2A–C). We used the mouse antisera for immunohistochemistry in cerebellar sections from a *C9orf72* patient and a healthy control (Fig 1C). The Ova-$(GA)_{10}$ antiserum specifically detected the pathognomonic dot-like inclusions in the molecular layer of the cerebellum (Fig 1C and D). We confirmed the specificity of Ova-$(GA)_{10}$ antiserum by blocking experiment using recombinant GST-$(GA)_{15}$ (Fig 1E) and double immunofluorescence using a rabbit anti-GA antibody (Fig EV3). Thus, our prime-boosting regimen using Ova-$(GA)_{10}$ induces high levels of specific anti-GA antibodies without apparent side effects in mice.

### Ova-$(GA)_{10}$ immunization prevents motor deficits in GA-CFP mice by reducing poly-GA aggregation

GA-CFP mice develop motor and coordination deficits starting at the age of 4 months (Schludi *et al*, 2017). To analyze the therapeutic effect of immunization, we assessed motor performance using weekly beam walk assay starting at 9 weeks of age before the onset of symptoms. From week 15, control-immunized GA-CFP mice (TG-PBS) needed significantly more time than wild-type mice to cross the beam consistent with our previous findings (Figs 2A and EV4). Occasionally, TG-PBS mice could not complete the beam walk task and dropped down, which was not observed in wild-type mice and hardly occurred in TG-Ova-$(GA)_{10}$ mice (Fig 2B). Consistent with the lack of antibody response, $(GA)_{15}$-immunized GA-CFP mice (TG-$(GA)_{15}$) performed as poor as TG-PBS mice. In contrast, Ova-$(GA)_{10}$-immunized GA-CFP mice (TG-Ova-$(GA)_{10}$) initially developed motor deficits compared to wild-type littermates at week 17, but improved at later time points and performed close to wild-type level suggesting that a high antibody titer is required for the beneficial effect of immunization. Immunohistochemistry of the spinal cord confirmed the typical neuronal cytoplasmic poly-GA inclusion pathology in GA-CFP transgenic mice (Fig 2C; Schludi *et al*, 2017).

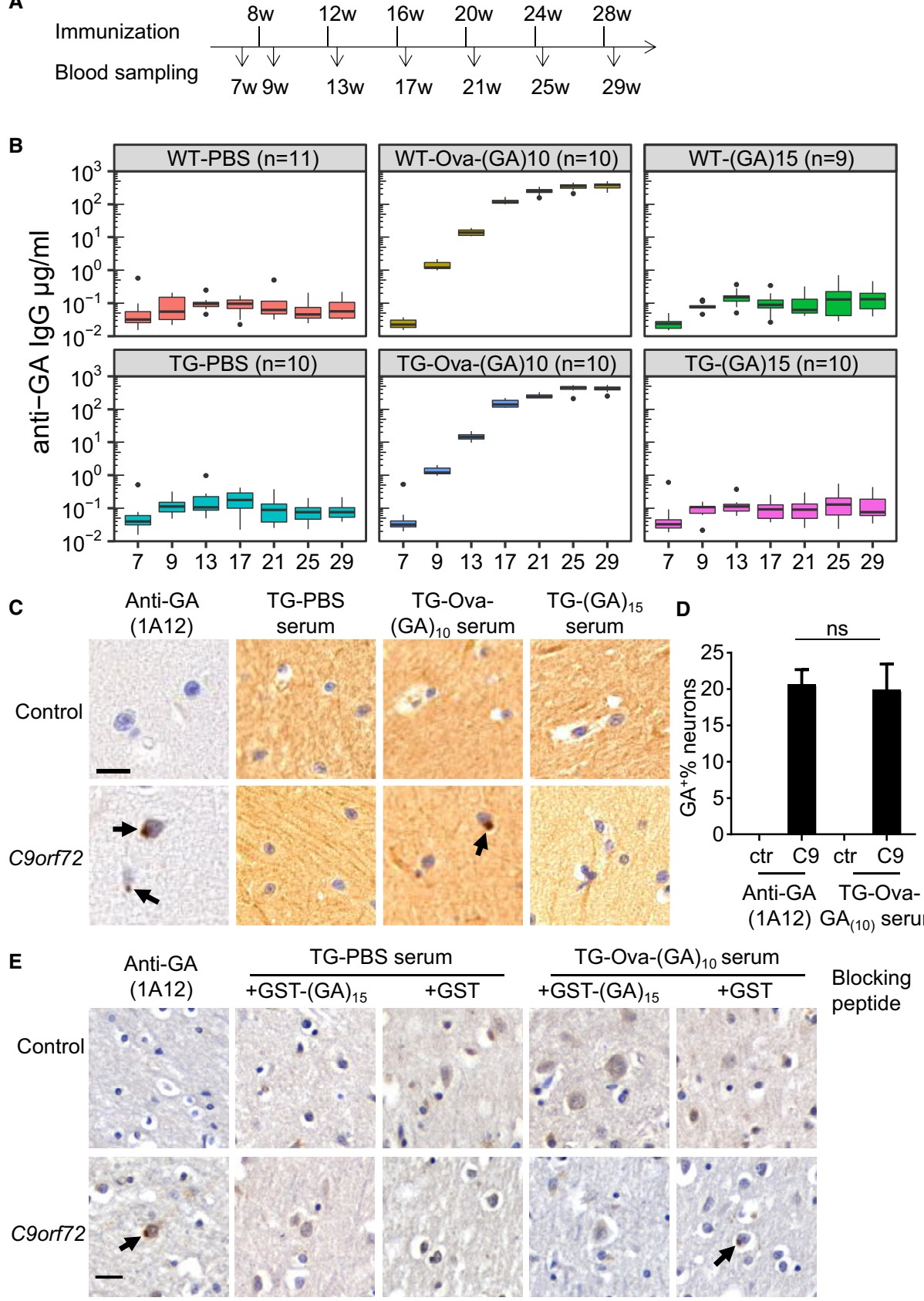

Figure 1.

◀

**Figure 1. Vaccination of carrier-coupled poly-GA peptides induces high-titer anti-GA antibodies.**

A  Experimental paradigm for vaccination of GA-CFP mice and wild-type littermates.

B  Anti-GA response in vaccinated mice measured by ELISA using GST-$(GA)_{15}$ antigen. Tukey-style box plot shows $25^{th}$, $50^{th}$, and $75^{th}$ percentiles, and whiskers extend to $\pm$ 1.5 interquartile range. Outliers depicted as dots. Number of mice per group as indicated. Two-way repeated-measure ANOVA (group x time) revealed a significant main effect of treatment ($F_{5,57}$ = 478.2, $P < 0.0001$) and time ($F_{6,342}$ = 296.1, $P < 0.0001$) and a significant interaction between factors ($F_{30,342}$ = 119.5, $P < 0.0001$), followed by Tukey's *post hoc* test. *P*-values are indicated in Appendix Table S1.

C  Immunohistochemistry of sections from a *C9orf72* patient and a healthy control. Monoclonal anti-GA and antiserum from Ova-GA-vaccinated mice detect neuronal cytoplasmic inclusions (arrows) in the molecular layer of the cerebellum. Note that crude antiserum shows higher background than the monoclonal antibody. Scale bar indicates 20 μm.

D  Quantitative analysis for poly-GA-positive inclusions per 100 neurons in (C). Bar graphs represents mean $\pm$ SD from 6 images. One-way ANOVA, Tukey's *post hoc* test, $F_{3,15}$ = 2.676, $P = 0.0846$, C9: anti-GA 1A12 vs. C9: TG-Ova-$(GA)_{10}$ serum $P = 0.9483$.

E  Immunohistochemistry of sections from occipital cortex of a *C9orf72* patient and a healthy control with monoclonal anti-GA and antiserum preincubated with 0.1 mg/ml recombinant GST-$(GA)_{15}$ or GST. Scale bar indicates 20 μm. Arrows depict poly-GA positive inclusions.

Image-based quantification revealed reduced aggregate density in TG-Ova-$(GA)_{10}$ mice compared to TG-PBS control (Fig 2D). This was further confirmed using a poly-GA immunoassay (Fig 2E). Consistent with the lack of antibody response and lack of any beneficial effect in the beam walk, $(GA)_{15}$ immunization did not affect the abundance of poly-GA aggregates in both assays. Taken together, repeated Ova-$(GA)_{10}$ immunization is an effective prevention strategy in GA-CFP mice.

### Ova-$(GA)_{10}$ immunization reduced neuronal damage, TDP-43 mislocalization, and microglia/macrophage activation in GA-CFP mice

To elucidate the mode of action, we performed unbiased RNA sequencing analysis on spinal cord from TG-PBS, TG-Ova-$(GA)_{10}$ mice, and their respective wild-type control mice. Pairwise comparison revealed no significant difference between the wild-type groups, but 545 differentially expressed genes between TG-PBS vs. WT-PBS and 233 between TG-Ova-$(GA)_{10}$ vs. TG-PBS using 1.5-fold change ($\log_2 > 0.585$) as cutoff (Datasets EV1–EV4). 210 genes differentially expressed in TG-PBS were significantly rescued in the immunized mice. Poly-GA expression triggered many immune pathways including production of multiple cytokines/chemokines. Ova-$(GA)_{10}$ immunization attenuated several immune pathways, for example, induction of *Ccl4*, *Grn*, *Tyrobp*, and complement factors (Fig 3A and B, and Dataset EV1). Thus, we analyzed activated microglia/macrophages using Iba1 immunohistochemistry. Similar to our previous report, GA-CFP transgenic mice showed strong microglia/macrophage activation in the spinal cord compared to wild-type littermates (Fig 3C). Both the density of Iba1-positive microglia/macrophages and the area covered by Iba1 staining were strongly reduced in TG-Ova-$(GA)_{10}$ mice (Figs 3C and D, and EV5A). Correspondingly, automated analysis of microglia/macrophage 3D morphology revealed significantly altered key features of microglial morphology (sphericity, number of branches) in TG-PBS mice compared to WT mice, which was significantly rescued in TG-Ova-$(GA)_{10}$ mice (Fig EV5B–D).

Since poly-GA triggers modest TDP-43 phosphorylation and partial cytoplasmic mislocalization of TDP-43 (Khosravi *et al*, 2017; Lee *et al*, 2017; Schludi *et al*, 2017; Nonaka *et al*, 2018), we next analyzed the effect of immunization on the levels of cytoplasmic TDP-43 in the spinal cord (Figs 3E and EV5E). While TG-PBS showed a significant increase in cells with cytoplasmic TDP-43 compared to control animals, mice immunized with Ova-$(GA)_{10}$ but

not $(GA)_{15}$ showed reduced levels of cytoplasmic TDP-43 compared to TG-PBS suggesting anti-GA antibodies reduce secondary TDP-43 pathology.

Finally, we analyzed the level of neurofilament light chain (NFL), which is as a biomarker for neuroaxonal damage in ALS and other diseases (Meeter *et al*, 2016; Feneberg *et al*, 2018; Khalil *et al*, 2018). Indeed, we observed significantly reduced level of NFL in CSF of TG-Ova-$(GA)_{10}$ suggesting the attenuation of neuroaxonal damage in these mice (Fig 3F). Taken together, Ova-$(GA)_{10}$ vaccination partially prevented neurodegeneration, reducing TDP-43 mislocalization and microglia/macrophage activation in poly-GA mice.

## Discussion

We show that poly-GA vaccination is safe and effective in a *C9orf72* mouse model. An immunogenic carrier protein such as ovalbumin can greatly enhance immunogenicity of poly-GA and can drive a high-level antibody response that would be difficult to maintain with regular intravenous injection of monoclonal antibodies. Presymptomatic poly-GA vaccination reduces inclusions and largely prevents TDP-43 mislocalization, neuroinflammation, neuroaxonal damage, and motor deficits in GA-CFP mice suggesting vaccination is a promising prevention strategy in the long prodromal phase of *C9orf72* ALS/FTD.

### Antigenicity and side effects

The humoral immune system has evolved to combat viruses and bacteria and strongly responds to other similar sized particles. This may explain why vaccination with aggregated full-length Aβ without carrier protein in the AN-1792 formulation was highly immunogenic (Schenk *et al*, 1999; Lee *et al*, 2005). We tested the antibody response using pre-aggregated $(GA)_{15}$, which would avoid use of carrier proteins that may cause unwanted side effects. However, even repeated boosting did not trigger any antibody response and showed no beneficial effects on motor function and histopathology. Since previous immunizations with soluble GST-$(GA)_{15}$ antigen resulted in only very weak antibody responses in rabbits (Mori *et al*, 2013), we covalently coupled $(GA)_{10}$ to maleimide-activated ovalbumin resulting in a multivalent antigen (5:1–15:1 ratio) that we also used to generate monoclonal antibodies (Mackenzie *et al*, 2013). Indeed, immunization with Ova-$(GA)_{10}$

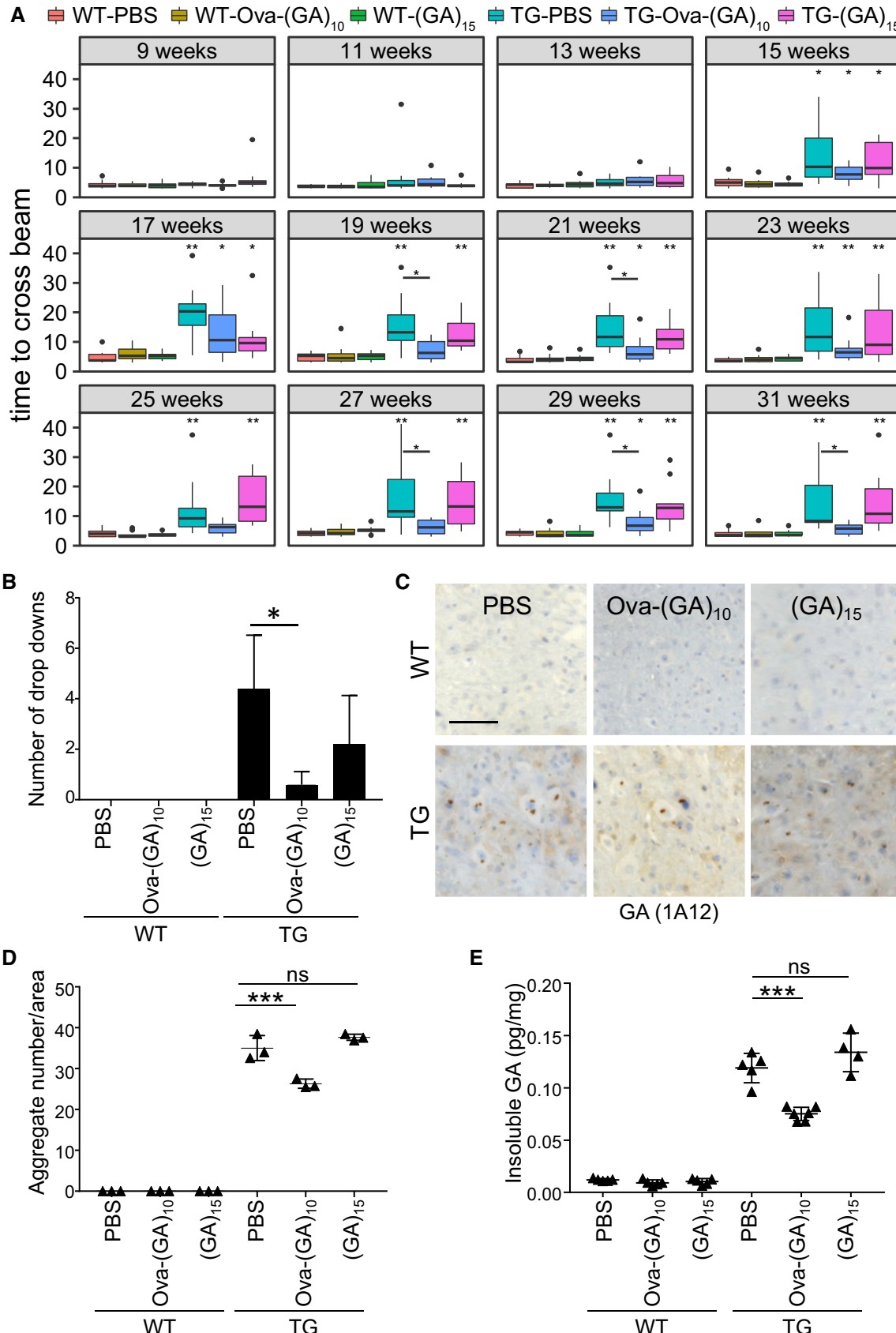

**Figure 2.**

**Figure 2. Ova-GA immunization prevents motor deficits and reduces poly-GA aggregation.**

A   Analysis of motor function in vaccinated GA-CFP mice and wild-type littermates in a beam walk assay. Average time to cross the beam from duplicate repeat measurements in consecutive weeks. Tukey-style box plot shows 25th, 50th, and 75th percentiles, and whiskers extend to ± 1.5 interquartile range. Outliers depicted as dots. Pairwise Wilcoxon rank sum test with Benjamini–Hochberg correction. All comparisons with WT-PBS and the TG-Ova-(GA)$_{10}$ mice vs. TG-PBS comparison are depicted with $*P < 0.05$, $**P < 0.01$. Number of mice per group as in Fig 1B, exact $P$-values are indicated in Appendix Table S2.

B   Average number of drop-downs per mouse from all aggregated runs. Number of mice per group as in Fig 1B. Bar graph represents mean ± SD. Kruskal–Wallis test with Dunn's multiple comparisons. $*P < 0.05$. TG-Ova-(GA)$_{10}$ vs. TG-PBS $P = 0.0188$.

C   Representative image of poly-GA aggregates in the anterior horn in spinal cord using immunohistochemistry. Scale bar indicates 100 μm.

D   Quantitative analysis of poly-GA aggregates in the spinal cord using immunohistochemistry. Images from complete spinal cord sections at 1-mm intervals from $n = 3$ animals. Dot plot represents mean ± SD. One-way ANOVA, Tukey's *post hoc* test, $F_{5,12} = 1.625$, $P = 0.2270$. $***P < 0.001$, ns not significant. TG-Ova-(GA)$_{10}$ vs. TG-PBS $P < 0.0001$; TG-(GA)$_{15}$ vs. TG-PBS $P = 0.2460$.

E   Immunoassay of poly-GA from the insoluble fraction of spinal cord lysate. Dot plot represents mean ± SD from $n = 5$-6 animals per group. One-way ANOVA, Tukey's *post hoc* test, $F_{5,24} = 3.098$, $P = 0.0269$. $**P < 0.001$, ns not significant. TG-Ova-(GA)$_{10}$ vs. TG-PBS $P < 0.0001$; TG-(GA)$_{15}$ vs. TG-PBS $P = 0.1975$.

Source data are available online for this figure.

using an extended boosting regimen resulted in high antibody titers in GA-CFP as well as wild-type mice.

The high antibody titers of ~ 400 μg/ml would be difficult to maintain over prolonged time periods using intravenous injection of monoclonal antibodies, which typically results in 100–200 μg/ml serum levels that drop sharply within weeks in humans (Sevigny *et al*, 2016; Landen *et al*, 2017). Low antibody trough levels and development of anti-drug antibodies are known to limit the long-term efficacy of antibody therapy in oncology and inflammatory diseases (Mazor *et al*, 2014; Kverneland *et al*, 2018). For neurodegenerative disease, the poor delivery across the blood–brain barrier is rate-limiting. Antibody engineering can increase antibody delivery from ~ 0.05 to 0.5% (Yu *et al*, 2014). Thus, maintaining high antibody levels is likely superior to intermittent i.v. antibody administration even using the best available methods to enhance blood–brain barrier delivery.

The safety profile of life-long administration of highly immunogenic carriers will need to be investigated before widespread application. We noticed no adverse effects such as T-cell infiltration or other signs of meningoencephalitis, but unfortunately experiments in mice cannot sufficiently predict the T-cell response in humans (Schenk *et al*, 1999; Orgogozo *et al*, 2003).

### Mode of action

So far, active vaccination in neurodegenerative diseases has often served as an entry point for passive immunotherapy, but given a good safety profile, active vaccination would be ideal for prevention. Benefits of preventive vaccination have been first reported in

Aβ mouse models (Schenk *et al*, 1999) and later in α-synuclein mice (Masliah *et al*, 2005) and Tau transgenic mice (Asuni *et al*, 2007; Boimel *et al*, 2010) and rats (Kontsekova *et al*, 2014). In contrast to the side effects of Aβ vaccination in humans, repeated active immunization with a peptide vaccine targeting Tau showed no adverse events in a phase 1 trials (Novak *et al*, 2017, 2018b). Cytoplasmic aggregates of Tau and α-synuclein have also been successfully targeted by antibodies, but the mode of action is still under debate (Masliah *et al*, 2011; Yanamandra *et al*, 2013; Games *et al*, 2014; El-Agnaf *et al*, 2017; Congdon & Sigurdsson, 2018). Antibodies may prevent seeding and cell-to-cell transmission as shown for Tau (Rajamohamedsait *et al*, 2017; Albert *et al*, 2019).

Our data that anti-GA vaccination predominantly reduces microglia-mediated neuroinflammation suggest that the resulting antibodies may bind and neutralize secreted poly-GA. We had previously shown that monoclonal anti-GA antibodies can reduce poly-GA uptake and seeding of patient brain extracts (Zhou *et al*, 2017). Alternatively, antibody uptake may enhance lysosomal or proteasomal degradation, for example, via the intracellular Fc-receptor TRIM21 (Mallery *et al*, 2010; McEwan *et al*, 2017), which may reduce neuronal stress signals that trigger harmful microglia/macrophage activation. Our transcriptome analysis confirms that transgenic poly-GA expression induces neuroinflammation and Ova-(GA)$_{10}$ immunization attenuated several immune pathways. Our finding that anti-GA vaccination also rescues the level of cytoplasmic TPD-43 suggests that targeting a single component may be sufficient to delay or even stop disease progression in *C9orf72*-associated ALS/FTD. Importantly, TDP-43 aggregation has also been linked to microglia activation (Swarup *et al*, 2011; Zhao *et al*, 2015).

**Figure 3. Ova-GA immunization prevents microglia/macrophage activation, TDP-43 mislocalization, and neuroaxonal damage.**

A   Gene ontology analysis of differentially expressed genes in TG-PBS mice comparing genes significantly rescued by Ova-(GA)$_{10}$ immunization and non-rescued genes (absolute log2-fold change > 0.585, compare Datasets EV1–EV4). The dot size and color represent the fraction of the differentially expressed genes in each category and adjusted $P$-values, respectively. Mouse number as indicated in Fig 1B.

B   Network of the genes dysregulated in TG-PBS and significantly rescued in TG-Ova-(GA)$_{10}$.

C, D   Analysis of microglia/macrophage activation using Iba1 immunohistochemistry from complete spinal cord sections at 1-mm interval. Dot plot represents mean ± SD from $n = 3$ animals per group. One-way ANOVA, Tukey's *post hoc* test. $**P < 0.01$, $*P < 0.05$, ns not significant. Scale bar indicates 100 μm. $F_{5,12} = 0.6974$, $P = 0.6357$; TG-Ova-(GA)$_{10}$ vs. TG-PBS $P = 0.0131$; TG-(GA)$_{15}$ vs. TG-PBS $P = 0.7650$. TG-PBS vs. WT-PBS $P = 0.0018$.

E   The percentage of cells with partial cytoplasmic mislocalization of TDP-43 (compare Fig EV5E) was quantified from six images from spinal cord sections at 1-mm intervals. Dot plot represents mean ± SD from $n = 3$ animals per group. One-way ANOVA, Tukey's *post hoc* test. $**P < 0.01$, $***P < 0.001$, ns not significant. $F_{5,12} = 0.6533$, $P = 0.6650$; TG-Ova-(GA)$_{10}$ vs. TG-PBS $P = 0.0096$; TG-(GA)$_{15}$ vs. TG-PBS $P = 0.4215$.

F   Immunoassay of NFL level in cerebrospinal fluid. Dot plot represents mean ± SD from $n = 4$ animals per group. One-way ANOVA, Tukey's *post hoc* test, $F_{3,11} = 1.911$, $P = 0.1862$. $**P < 0.01$. TG-Ova-(GA)$_{10}$ vs. TG-PBS $P = 0.0081$.

Source data are available online for this figure.

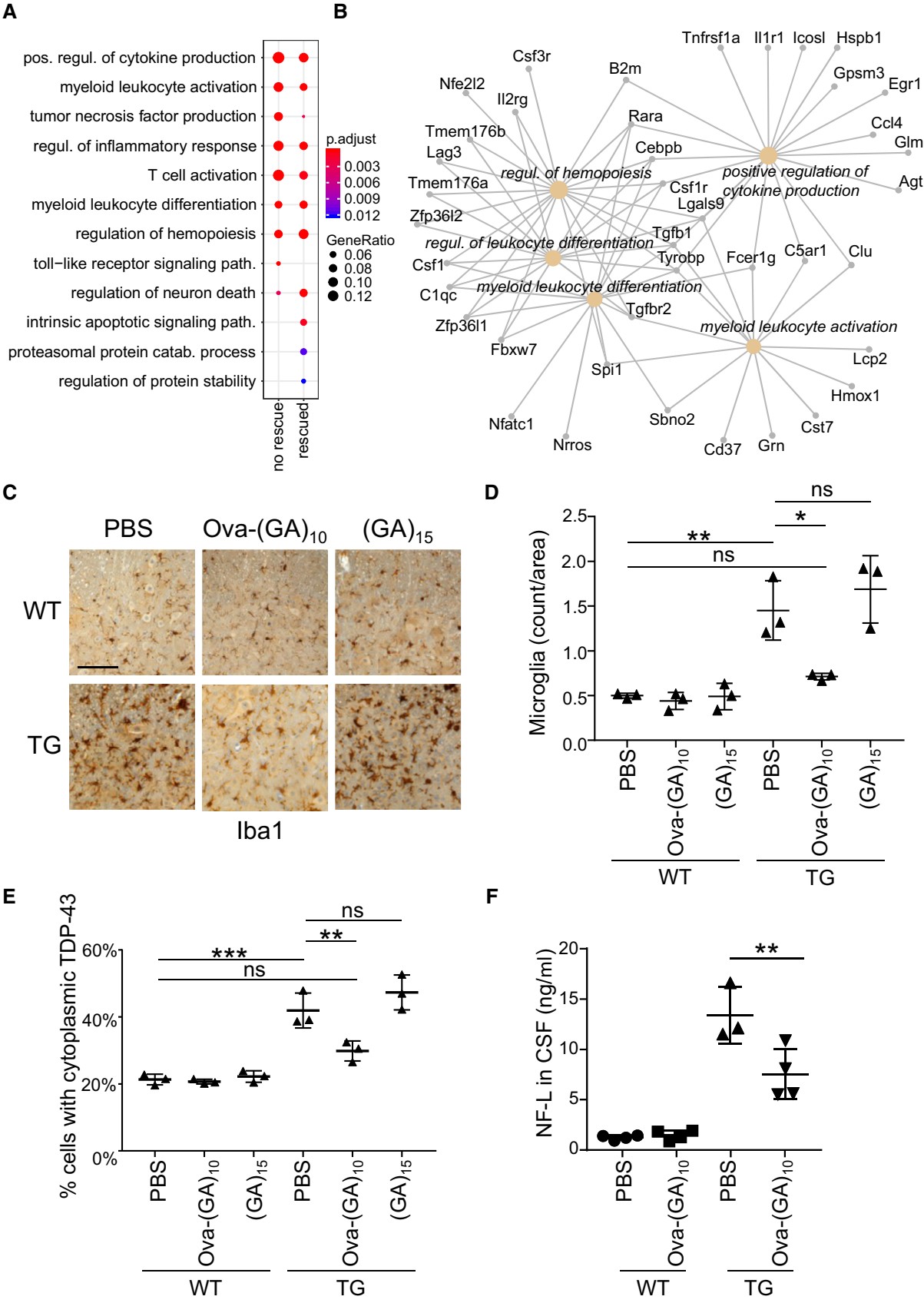

**Figure 3.**

### Prevention vs. therapy

Familial variants of neurodegenerative diseases offer the unique chance for true disease prevention until somatic gene editing may allow a genetic cure in the future. With the rise of genetic testing in ALS/FTD patients, an increasing number of relatives become aware of their high disease risk and are eagerly looking for therapeutic options. Preventive treatment with antisense oligonucleotides ameliorated disease in *C9orf72* mouse models but would require regular intrathecal injection (Jiang *et al*, 2016; Gendron *et al*, 2017). Moreover, reaching a sufficient dose throughout the human brain will be far more difficult than targeted therapy of the spinal cord in spinal muscular atrophy (Chiriboga *et al*, 2016). Although current poly-GA models suggest that poly-GA is not sufficient to mimic the complete pathophysiology of ALS/FTD in mice (Zhang *et al*, 2016; Schludi *et al*, 2017), targeting poly-GA may be sufficient to prevent ALS/FTD, which should be tested in *C9orf72* BAC mice in the future (Jiang *et al*, 2016; Liu *et al*, 2016). Importantly, immunotherapy reduces poly-GA-dependent TDP-43 pathology in our mouse model. Taken together, active vaccination is a promising approach to reduce disease severity or even prevent ALS and FTD in people carrying the *C9orf72* mutation and highlights the importance poly-GA-induced neuronal damage and microglia/macrophage activation.

## Materials and Methods

### Mice and immunization

$(GA)_{149}$-CFP-expressing transgenic (TG) mice from C57BL/6J background have been characterized in detail (Schludi *et al*, 2017). For genotyping, mice genomic DNA was prepared using Hot Sodium Hydroxide and Tris (HotSHOT) method as described previously (Truett *et al*, 2000). Genotyping PCR was performed using the primers 5′-tccaggagcgtaccatcttc-3′ and 5′-gtgctcaggtagtggttgtc-3′. The presence of the full-length transgene was confirmed with PCR amplification (Expand Long Template PCR System, Roche, 5′-gatcc aagcttgccaccatg-3′ and 5′-tctagctctgccactccaag-3′) and sequencing.

Immunization followed the protocol for generation of mouse monoclonal anti-GA antibodies (Mackenzie *et al*, 2013) with additional boosting immunizations (prime-boost regimen as in Fig 1A). For the first immunization, 40 µg Ova-$(PEG)_3$-$(GA)_{10}$ or $(GA)_{15}$ (Peptide Specialty Laboratories GmbH, Heidelberg) was mixed with 5 nmol CpG ODN 1668 oligonucleotide (Enzo Life Science) in 200 µl PBS and 250 µl incomplete Freund's adjuvant and injected half-half intraperitoneally (i.p.) and subcutaneously (s.c.) at 8 weeks of age. The control group (PBS) received injection of 200 µl PBS and 250 µl incomplete Freund's adjuvants together with 5 nmol CpG ODN 1668. For the five consecutive booster immunizations, 40 µg Ova-$(PEG)_3$-$(GA)_{10}$ or $(GA)_{15}$ was mixed with 5 nmol CpG ODN 1668 in 500 µl PBS and was injected half-half i.p. and s.c. at weeks 12, 16, 20, 24, and 28 (see Fig 1A). The control group (PBS) received injection of 495 µl PBS together with 5 nmol CpG ODN 1668. Nine to 11 animals with similar gender ratio per group were included in the study.

Animal handing and animal experiments were performed in accordance with institutional guidelines approved by local animal laws and housed in standard cages in pathogen-free facility on a 12-h light/dark cycle with *ad libitum* access to food and water.

### Beam walk test

Mice were placed on a wooden beam (round surface, length 58 cm, diameter 8 mm) and had 1 min to cross the beam. The measurement was completed when the mice reached the end of the stick, they dropped down, or the time ran out. For the statistical analysis, failed attempts were set to 1 min. The experimenters were blind to the genotype, and trials were video-documented for the analysis. Starting at the age of 9 weeks old, the assay was repeated weekly with two consecutive trials. Results from two consecutive weeks (four total runs) were averaged for analysis.

### Serum and CSF collection and tissue harvesting

At 7, 9, 13, 17, 21, 25, and 29 weeks of age, blood was taken and incubated 15 min at room temperature. Serum was collected by centrifugation (13,000 *g* for 10 min at 4°C) and snap-frozen on dry ice. After the behavior experiments, animals were anesthetized by intraperitoneal administration of medetomidine (0.5 mg/kg) + midazolam (5 mg/kg) + fentanyl (0.05 mg/kg). Cerebrospinal fluid (CSF) was collected from the cisterna magna according to the previously published methods (Schelle *et al*, 2017). Transcardial perfusion was performed with cold phosphate-buffered saline (PBS). Tissue was either stored at −80°C for biochemical analysis or formalin-fixed for 24 h for immunohistochemistry analysis. For spinal cord tissue, an additional decalcification step with 5% formic acid for 48 h after formalin fixation was performed. For the microglia morphology study, thoracic segments T1–T7 were dissected after formalin fixation.

### Primary antibodies

Antibodies to poly-GA (1:500, Proteintech, 24492-1-AP), GFP (1:1,000, Clontech, 632592), Iba1 (1:500, Wako, 091-19741), TDP-43 (1:500, Proteintech, 10782-2-AP), calnexin (1:3,000, Enzo Life Science, SPA-860F), CD3 (eBioscience, 17A2), anti-CD4 (clone: RM4-5, eBioscience), CD8 (eBioscience, 53-6.7), CD19 (eBioscience, eBio1D3), CD45 (eBioscience, 30-F11), and CD11b (eBioscience, M1/70) are commercially available. In addition to our previous anti-GA clone 5F2 (purified mouse monoclonal, biotinylated 1:1,000, Sulfo-Tagged 10 ng/µl) used for immunoassay, we used clone 1A12 established from the same immunization (Mackenzie *et al*, 2013), which shows superior staining specificity (mouse monoclonal, IgG1, WB 1:50, IHC 1:100).

### Antibody titer ELISA

To measure the antibody titer, Nunc MaxiSorp™ flat-bottom 96-well plates were incubated with recombinant GST-$(GA)_{15}$ produced in *E. coli* (Mori *et al*, 2013) in PBS overnight at 4°C. After 1 h incubation in blocking solution (1% BSA, 0.05% Tween 20 in PBS), samples (predilute 1:100 or 1:100,000) were added for 1 h. After three washes with 0.05% Tween 20 in PBS, anti-GA antibodies were detected with anti-mouse HRP. 3,3′,5,5′-Tetramethylbenzidine (Sigma) was used as chromogenic substrate. After stopping the

reaction with 2M $H_2SO_4$, the absorbance was read at 450 nm. Serial dilution of purified mouse monoclonal anti-GA clone 1A12 was used to generate a standard curve for absolute quantification. Two technical replicates were measured for each sample.

### DNA constructs, Cell culture, transfection, immunofluorescence, and immunoblot

ATG-initiated epitope-tagged synthetic expression constructs for $GA_{175}$-GFP, $GA_{149}$-myc, $GP_{47}$-GFP, GFP-$GR_{149}$, $PA_{175}$-GFP, and $PR_{175}$-GFP in pEF6/V5-His vector were described previously (May *et al*, 2014; Schludi *et al*, 2015; Zhou *et al*, 2017).

HEK293FT cells were transfected using Lipofectamine 2000 (Thermo Scientific) according to the manufacturer's instruction and cultivated with DMEM containing 10% FCS and penicillin/streptomycin. For immunofluorescence, cells were fixed with 4% paraformaldehyde and 4% sucrose for 10 min and stained with the indicated antibodies in GDB buffer (0.1% gelatin, 0.3% Triton X-100, 450 mM NaCl, 16 mM sodium phosphate pH 7.4). Images were taken using an LSM810 confocal laser scanning system (Carl Zeiss) with 40× or 63× oil immersion objectives.

For immunoblotting, samples were prepared as described previously (May *et al*, 2014) and separated on 10–20% tricine protein gels (Thermo Fisher) and transferred to PVDF membranes. Membranes were blocked with 0.02% I-block in PBS with 0.2% Triton X-100 for 1 h and then incubated with the respective antibodies overnight at 4°C. After washing with TBST, membranes were incubated with anti-mouse HRP antibody for 1 h at RT. After washing with TBST, signals were developed with ECL (Thermo) and detected with Super RX Fuji medical X-Ray film (Fujifilm).

### Transcriptome analysis

We isolated RNA from thoracic segments T8–T13 of spinal cord from WT and GA-CFP mice immunized with PBS or OVA-$(GA)_{10}$ using Direct-zol RNA MicroPrep (Zymo Research), generated libraries using the mRNA SENSE kit V2 from Lexogen (Vienna, Austria) essentially according to the manufacturer's instructions and performed 100 bp single-read sequencing on a Illumina HiSeq1500 (Illumina, San Diego CA, USA). Reads were aligned to the mouse genome (mm10) using STAR (Version 2.5.2b-0) (Dobin *et al*, 2013) supplied with the UCSC gene models obtained from iGenomes. Count tables were generated using HTSeq-count (Version 1.0.0), and differential expression for WT-OVA vs. WT-PBS, TG-PBS vs. WT-PBS, and TG-OVA vs. TG-PBS was performed using DESeq2 (Version 2.11.40.6) (Love *et al*, 2014) in Galaxy (full expression analysis is presented in Datasets EV2–EV4; Afgan *et al*, 2018). Genes with adjusted $P$-values below 0.05 and absolute $\log_2$-fold change higher than 0.585 (1.5-fold change) were used for gene ontology analysis using clusterProfiler (version 3.12) (Yu *et al*, 2012). A manual selection of meaningful and highly significantly enriched GO categories is shown in Fig 3A. The full data are presented in Dataset EV1.

### Flow cytometry analysis

Fresh spleens were transferred to Hank's balanced salt solution, homogenized and filtered through 40-μm cell strainers.

Erythrocytes in spleens were lysed using isotonic ammonium chloride buffer. Splenocytes were stained according to the manufacturer's protocol and were measured on a BD FACSverse flow cytometer (BD Biosciences) and analyzed using FlowJo 10.4 software (Treestar).

### Poly-GA immunoassay

The poly-GA immunoassay was performed to measure insoluble poly-GA in the spinal cord of GA-CFP mice as described previously with modifications (Schludi *et al*, 2017). Mouse spinal cord samples were homogenized in 1 ml of RIPA buffer (137 mM NaCl, 20 mM Tris pH 7.5, 10% glycine, 1% Triton X-100, 0.5% Na-deoxcholate, 0.1% SDS, 2 mM EDTA, protease and phosphatase inhibitors, benzonase nuclease) using homogenizer (Precellys). Samples were incubated with shaking at 4°C for 20 min and centrifuged at 13,000 $g$ for 10 min at 4°C. To avoid cross-contamination between soluble and insoluble fractions, pellets were resuspended in 1 ml RIPA, re-homogenized, and re-centrifuged. The RIPA-insoluble pellets were sonicated in RIPA buffer containing 3.5 M Urea (U-RIPA), and protein concentration was determined by Bradford assay. Streptavidin Gold multi-array 96-well plates were incubated with biotinylated anti-GA clone 5F2 overnight at 4°C and blocked with 1% BSA, 0.05% Tween 20 in PBS for 1 h. Equal amounts of samples were added in duplicate wells for 1.5 h, followed by 1.5 h incubation with the secondary MSD-labeled α-GA clone 5F2. Serial dilution of recombinant GST-$(GA)_{15}$ in blocking buffer was used to generate a standard curve. The intensity of emitted light upon electrochemical stimulation was measured using the MSD QuickPlex 520, and the background was corrected by the average response obtained from blank wells.

### Microglia morphology analysis

Automated morphological analysis of microglia in spinal cord was reported previously (Heindl *et al*, 2018). Free floating staining of 100-μm-thick vibratome sections of spinal cord were blocked and incubated with anti-Iba1 for 36 h. After intensive wash, sections were incubated with secondary antibody anti-rabbit coupled to Alexa Fluor 594. Nuclei were stained with DAPI. Sections were mounted with VECTASHIELD® Hardset™ Antifade Mounting Medium. Images were acquired using a Zeiss confocal microscope with 40× magnification (objective: EC Plan-Neofluar 40×/1.30 Oil DIC M27) with an image matrix of 1,024 × 1,024 pixel, a pixel scaling of 0.2 × 0.2 μm, and a depth of 8 bit. Confocal images were collected in $Z$-stacks with a slice distance of 0.4 μm. Images were processed using a MATLAB implemented processing pipeline as described previously.

### Patient tissue, immunohistochemistry, and quantitative analysis

All patient materials were provided by the Neurobiobank Munich and Ludwig-Maximilians-University (LMU) Munich and approved by the ethical committee of Ludwig-Maximilians-University (LMU) Munich. All subjects and the experiments were confirmed to the principles set out in the WMA Declaration of Helsinki and the Department of Health and Human Services Belmont Report. Informed consent was obtained from all subjects.

## The paper explained

### Problem

The poor results of clinical trials in amyotrophic lateral sclerosis (ALS) and other neurodegenerative disease suggest we still lack a basic understanding of causality. The key disease mechanisms may change during disease progression, and halting further neuron loss has not been achieved and may not even improve the symptoms and survival. About 10–20% of patients with ALS and 40–50% of patients' frontotemporal dementia (FTD) show a Mendelian pattern of inheritance and with the advent of genetic testing of patients many relatives at risk ask for preventive options. The most common known mutation found in about ~ 10% of patients in Europe and the United States is a $(G_4C_2)_n$ repeat expansion in the non-coding region of *C9orf72*. Sense and antisense repeat transcripts are translated in all reading frames into five aggregating dipeptide repeat (DPR) proteins that form predominantly neuronal cytoplasmic inclusions that are detectable many years prior to disease onset. In animal models, gain-of-function mechanisms are clearly the main drivers of pathogenesis, but it is unclear what ultimately triggers symptomatic disease characterized by regional TDP-43 pathology and neurodegeneration. We have previously shown that DPRs are transmitted between cells and asked whether antibody therapy that potentially target extracellular DPR species could be used for prevention or therapy of *C9orf72* ALS/FTD.

### Results

We immunized a transgenic mouse model expressing the most abundant DPR protein, poly-GA, using two different vaccine formulations to induce anti-GA antibodies *in vivo*. Only ovalbumin-coupled $(GA)_{10}$ but not $(GA)_{15}$ induced a high-titer antibody response (> 400 µg/ml) in a prime/boost regimen without side effects. Treated mice showed less poly-GA aggregation and cytoplasmic mislocalization of TDP-43. Importantly, vaccination with Ova-$(GA)_{10}$ largely prevented motor symptoms and microglia/macrophage activation *in vivo*. In addition, TDP-43 mislocalization and levels of neurofilament light chain, a biomarker for neuroaxonal damage, were reduced by Ova-$(GA)_{10}$ immunization suggesting improved neuronal health.

### Impact

We show that poly-GA immunotherapy may prevent onset of ALS and FTD in *C9orf72* mutation carriers by inhibiting protein aggregation and microglia/macrophage activation. Coupling poly-GA peptides with a carrier protein greatly enhanced immunogenicity of poly-GA leading to very high antibody titers that would be difficult to maintain with regular i.v. administration. Early treatment of *C9orf72* mutation carriers with a safe and effective poly-GA vaccine may prevent transition from the prodromal DPR-dominated disease stage to the symptomatic disease associated with TDP-43 pathology. In addition, our data support development of passive immunotherapy with monoclonal antibodies.

Immunohistochemistry staining was performed on 5-µm-thick paraffin-embedded tissue sections. After deparaffinization and dehydration, heat-induced antigen retrieval was performed with citrate buffer (pH 6) for 20 min in the microwave. Afterward, the slides were blocked and incubated with primary antibody overnight at 4°C. The slides were washed and detected by the DCS supervision 2 Kit (DCS innovative diagnostic-system) according to the manufacturer's instructions. Iba1 staining was performed with the Ventana BenchMark XT automated staining system (Ventana) using the UltraView Universal DAB Detection Kit (Roche). Bright-field images were taken on Leica DMi8 fluorescence microscope (Leica). Peptide blocking assay was performed as described previously (Mori *et al*,

2013) by preincubating mouse serum with 0.1 µg/µl GST-$(GA)_{15}$ or GST at 4° for 1 h.

Quantification was performed using ImageJ software. For quantitative analysis of poly-GA aggregates in the spinal cord, images were analyzed from spinal cord sections at 1-mm intervals from $n = 3–4$ mice per group, and all positively stained cells were manually counted. For quantitative analysis of microglia number from immunohistochemistry stainings, the Image Analysis Toolbox was used according to the developer's instructions (Shu *et al*, 2013).

### Statistical analysis

Blood sampling and immunization were performed in a random order. Sample distribution for immunoassay was randomized. Application of antibodies for immunohistochemistry staining or immunofluorescent staining was performed in a randomized order. Investigators were blinded to genotype during data collection by the use of a number code for each animal or human subject.

Statistical analysis was performed using R and RStudio (R Core Team, 2018) or Prism 7 (GraphPad Software, Inc.). We used one-way ANOVA with Turkey correction to compare more than two groups. Data distributions were checked for normality by the Shapiro–Wilk test, and homogeneity of variances was checked by the F-test or the Brown–Forsythe test. When these were violated, non-parametric tests were used with Kruskal–Wallis test with Benjamini–Hochberg correction. Wilcoxon rank sum test with continuity correction was applied for group comparisons.

# Data availability

The datasets produced in this study are available in the following database: RNA-seq data: Gene Expression Omnibus GSE138413 (https://www.ncbi.nlm.nih.gov/geo/query/acc.cgi?acc = GSE138413).

**Expanded View** for this article is available online.

### Acknowledgements

We thank Bahram Khosravi, Kai Schlepckow, Sabina Tahirovic, and Xianyuan Xiang for critical comments to the manuscript. This work was supported by NOMIS Foundation and the Hans und Ilse Breuer Foundation (C.H. and D.E.), the Munich Cluster of Systems Neurology (SyNergy) (D.E., Q.Z., A.L., C.H. R.F.), and the European Community's Health Seventh Framework Programme under grant agreement 617198 (DPR-MODELS) (D.E.).

### Author contributions

QZ and NM designed, performed, and interpreted the immunization study with technical help from MM and DF. SP collected CSF samples from mice. BN, MC, and CS performed and interpreted the biomarker studies. CH supervised biomarker study and provided technical resources. SH, SR, and AL performed and interpreted the analysis of microglia morphology and helped with flow cytometry. AG, SK, and HB performed RNA-seq and helped with the data analysis. RF generated anti-GA clone 1A12. TA provided human tissue and supervised neuropathological analysis and interpreted results. DE designed and interpreted the study and helped with data analysis. DE and QZ wrote the manuscript with input from all authors.

## Conflict of Interest

Q.Z., N.M., and D.E. are inventors on a patent application filed for poly-GA vaccination targeting *C9orf72* ALS/FTD. C.H. is the chief consultant of ISAR Bioscience.

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
