## [Review Process File · EMBO Molecular Medicine]

Active poly-GA vaccination prevents microglia activation and motor deficits in a *C9orf72* mouse model

Qihui Zhou, Nikola Mareljic, Meike Michaelsen, Samira Parhizkar, Steffanie Heindl, Brigitte Nuscher, Daniel Farny, Mareike Czuppa, Carina Schludi, Alexander Graf, Stefan Krebs, Helmut Blum, Regina Feederle, Stefan Roth, Christian Haass, Thomas Arzberger, Arthur Liesz, Dieter Edbauer

Review timeline:

Submission date:	21 May 2019
Editorial Decision:	19 June 2019
Revision received:	11 October 2019
Editorial Decision:	11 November 2019
Revision received:	29 November 2019
Accepted:	3 December 2019

Editor: Céline Carret

Transaction Report:

1st Editorial Decision

19 June 2019

Thank you for the submission of your manuscript to EMBO Molecular Medicine. We have now heard back from the three referees whom we asked to evaluate your manuscript.

As you will see from the reports below, the referees find the topic of your study of potential interest. However, they raise substantial concerns on your work, which should be convincingly addressed in a major revision of the present manuscript. We would encourage you to revise the figures (ref. 1 and 3) and strengthen your findings with additional experiments as recommended by ref. 2 and 3. Further, unbiased analyses (ref. 2) and a more detailed analysis of vaccination effect in vivo (ref. 3) would be nice additions.

We would therefore welcome the submission of a revised version within three months for further consideration and would like to encourage you to address all the criticisms raised as suggested to improve conclusiveness and clarity. Please note that EMBO Molecular Medicine strongly supports a single round of revision and that, as acceptance or rejection of the manuscript will depend on another round of review, your responses should be as complete as possible.

Please also contact us as soon as possible if similar work is published elsewhere. If other work is published, we may not be able to extend the revision period beyond three months.

I look forward to receiving your revised manuscript.

***** Reviewer's comments *****

Referee #1 (Remarks for Author):

This study demonstrates that immunization of a transgenic mouse model expressing the most abundant DPR protein, with ovalbumin-coupled (GA)₁₀, induced a high titer antibody response and resulted in a reduction in poly-GA aggregation, and reduced motor symptoms and microglial activation.

The overall approach is interesting and has clinical implications. Yet, there are some key issues in the data that require attention:

Results

1. In Figure 1c, the data presented in a way that not convincing.
2. Based on Figure 2, it is not clear how the authors concluded that the treatment prevented motor deficits.
3. Figure 3 shows only Iba1 expression as a basis for the claim of microglial activation. This is not sufficient, as Iba1 is not only marker of activated microglia, but also stains monocyte-derived macrophages. Also, the phenotype of the microglia is more critical than whether they are activated.

Discussion:

The authors should tone down the conclusions with respect to the overall preventive effect. The results are impressive if the effect is only reduction of disease severity.

Referee #2 (Comments on Novelty/Model System for Author):

How relevant transgenic mouse models in FTLN is still a question? This model accumulates GA dipeptide repeats but not decreased C9orf72 expression. However, such model is not yet available.

Referee #2 (Remarks for Author):

The present work by Zhou and coll. explores the use of immunotherapy by using an active vaccination against GA dipeptide repeats in a mouse model of FTLN.

They have tested both GA dipeptide repeats coupled to ovalbumin and polyGA aggregates as immunogens. Only GA peptides coupled to the protein carrier lead to an immune response. Such vaccination shows improvement in motor behaviour and decreased insoluble GA dipeptide repeats in the mouse model. Even if this approach is new for dipeptide repeats, such vaccination has been widely used in neurodegenerative disorders for Aβ, α-synuclein, tau, etc... It is not original and mostly descriptive. It does not bring any insights on DPR toxicity or etiopathogenesis. With OMICS analyses, the authors may have uncovered new pathways, but it has not been explored. It is an interesting study which shows nice preliminary data, but the authors have to further explore their findings (effect on cell toxicity, pathways involved, transcriptomics analysis...). Microglia is an evidence but try unbiased analyses.

Minor comments: many references are incomplete

Regarding statistical analysis, why ANOVA and not non-parametric tests?

Referee #3 (Remarks for Author):

GGGGCC repeat expansion is the most common genetic cause of both ALS and FTD. One major pathogenic mechanism is the production of dipeptide repeat (DPR) proteins through unusual RAN translation, as first reported by Dr. Edbauer and his colleagues a few years ago (Science, 2013). Indeed, numerous studies in cell cultures and animal models have demonstrated the toxicity of different DPR proteins, such as polyGA, polyGR, polyPR etc. These findings suggest that targeting DPRs may be a promising therapeutic approach.

In this study, Zhou and colleagues use an active immunization approach to generate anti-GA antibodies in mice expressing (GA)₁₄₉-GFP. Remarkably, they found that microglia activation in the spinal cord was largely prevented, although the reduction in polyGA inclusions was modest. This prove-of-concept study should be published (as a Correspondence, since it has limited scope), because it is the first report demonstrating beneficial effects of DPR-specific antibodies in the

ALS/FTD field, although similar approaches have been used numerous times for other neurodegenerative diseases.

In this study, they used animals that overexpress polyGA. It is expected that targeting polyGA should have some beneficial effects since it is overexpressed polyGA that causes these ALS/FTD disease-like phenotypes. It would be nice if they could demonstrate a similar beneficial effect in C9ORF72 BAC transgenic mice, where different DPR proteins as well as repeat RNAs are co-expressed and it is unclear which molecule causes neurodegeneration, such as the one reported by Dr. Ranum that shows some strong disease phenotypes. However, it is not reasonable to demand the authors to perform this time-consuming experiment at this stage, which will significantly delay the publication of this prove-of-concept study. Maybe it is feasible if they could please show whether polyGA level in the CSF is reduced after immunization, if so, the result would further strengthen their conclusion that rescue of motor function and microglia activation is largely due to extracellular polyGA.

1. The first sentence in the abstract does not seem to belong there. The point is better made in the Introduction and Discussion.
2. In the Introduction section, "neuronal cytoplasmic inclusions" should be "neuronal inclusions" since both nuclear and cytoplasmic inclusions have been reported.
3. In Figure 1B, could the authors please use different colors for mice with different treatment? Since they are all black and white, it is not obvious to tell TG-Ova-GA10 from others.
4. Figure 1C only shows a few images. This reviewer would like to suggest including some sort of quantification, which should not be difficult to do. For instance, they could count the number of neurons with inclusions per 100 cells in both control and patient brains. Also, it would be more convincing if additional brain tissues are examined, since the antisera give rise to high background signals, and/or a GA peptide blocking experiment is carried out to demonstrate the specificity of the Ova-GA10 antiserum.
5. In Figure 2B, the statistical difference between PBS and GA15 should be presented in the panel as well.
6. In Figure 3D, it is surprising to see 20% of cells in control mice show TDP-43 cytoplasmic localization. Is this number consistent with what the authors published before? In TG mice, the number is about 40%, which seems to be quite high. As Zhou and colleagues stated here: "poly-GA triggers modest TDP-43 phosphorylation and partial mislocalization of TDP-43". In contrast, the images in Figure 3C show a much lower % of cells with cytoplasmic TDP-43.
7. On Page 5, what does "ADD significance!" mean? It seems to be an error.

1st Revision - authors' response

11 October 2019

Referee #1 (Remarks for Author):

This study demonstrates that immunization of a transgenic mouse model expressing the most abundant DPR protein, with ovalbumin-coupled (GA)10, induced a high titre antibody response and resulted in a reduction in poly-GA aggregation, and reduced motor symptoms and microglial activation.

The overall approach is interesting and has clinical implications. Yet, there are some key issues in the data that require attention:

Results

1. In Figure 1c, the data presented in a way that not convincing.

Fig. 1c was done without mouse-on-mouse blocking reagents. We performed additional peptide blocking experiments using better conditions resulting in lower background (new Fig 1E) and additional show colocalization data in the new Fig EV3.

2. Based on Figure 2, it is not clear how the authors concluded that the treatment prevented motor deficits.

We reformatted the figure 2A to show cross-sectional differences at each time point with improved statistics and addition show longitudinal presentation of the data in the new Fig. EV4. Together the figures clearly show impaired performance of TG mice in the beam walk assay that is partially

prevented by Ova-(GA)₁₀ immunization. Thus, our therapy clearly improved the motor function in TG mice dependent on development of a high anti-GA antibody titer.

3. Figure 3 shows only *Iba1* expression as a basis for the claim of microglial activation. This is not sufficient, as *Iba1* is not only marker of activated microglia, but also stains monocyte-derived macrophages. Also, the phenotype of the microglia is more critical than whether they are activated. We agree that *Iba1* cannot fully distinguish microglia and macrophages. Therefore, we changed the text to "microglia/macrophages" throughout the manuscript. To address the phenotype of microglia/macrophages, we now performed in depth automated morphological analysis (new Fig. EV5C/D), which confirm activation in TG-PBS mice and significant rescue upon Ova-(GA)₁₀ immunization. The results are corroborated by transcriptome data (new Fig. 3A/B) showing partial rescue of inflammatory gene expression changes.

Discussion:

The authors should tone down the conclusions with respect to the overall preventive effect. The results are impressive if the effect is only reduction of disease severity.

The reviewer is right that we did not completely prevent symptoms in GA-CFP mice. We changed the conclusion of the discussion to "active vaccination is a promising approach to reduce disease severity or even prevent ALS and FTD in people carrying the C9orf72 mutation" as requested.

Referee #2 (Comments on Novelty/Model System for Author):

How relevant transgenic mouse models in FTLN is still a question? This model accumulates GA dipeptide repeats but not decreased C9orf72 expression. However, such model is not yet available. We agree that this model replicated only a part of C9orf72 disease, but ample evidence suggests that poly-GA plays a critical role and is most closely linked to TDP-43 pathology (Khosravi et al, 2017; Nonaka et al, 2018, Lee et al, 2017). BAC transgenic mouse model replicating patient-length (G₄C₂)_n repeats which seem to be not completely genetically stable (<https://www.jax.org/strain/029099> and <https://www.jax.org/strain/029102>) or show no motor or cognitive phenotype (O'Rourke et al, Neuron 2015, Peters et al, Neuron 2015). Concomitant downregulation of C9orf72 expression in BAC transgenic mice has not been reported. We clearly state the limitations of our study in the revised discussion.

Referee #2 (Remarks for Author):

The present work by Zhou and coll. explores the use of immunotherapy by using an active vaccination against GA dipeptide repeats in a mouse model of FTLN.

They have tested both GA dipeptide repeats coupled to ovalbumin and polyGA aggregates as immunogens. Only GA peptides coupled to the protein carrier lead to an immune response. Such vaccination shows improvement in motor behaviour and decreased insoluble GA dipeptide repeats in the mouse model. Even if this approach is new for dipeptide repeats, such vaccination has been widely used in neurodegenerative disorders for Aβ, α-synuclein, tau, etc... It is not original and mostly descriptive. It does not bring any insights on DPR toxicity or etiopathogenesis. With OMICS analyses, the authors may have uncovered new pathways, but it has not been explored.

It is an interesting study which shows nice preliminary data, but the authors have to further explore their findings (effect on cell toxicity, pathways involved, transcriptomics analysis...). Microglia is an evidence but try unbiased analyses.

We greatly expanded our manuscript as suggested by this and other reviewers. We now include transcriptome data from a replication cohort (new Fig. 3A/B, new Table S3-S6). Transcriptome analysis shows that poly-GA expression results in prominent neuroinflammation. These changes are partially rescued by OVA-(GA)₁₀ immunization. This crucial new data is now shown in the new Fig. 3A/B and Tables S3-S6. Consistent with these findings we also show that immunization reduces neuroaxonal damage using NFL immunoassays in the CSF (new Fig. 3F). Moreover, detailed analysis of the microglia morphology reveals poly-GA induced neuroaxonal damage also confirms partial rescue by immunization (new Fig. EV5B-D).

Although immunotherapy has been attempted in other neurodegenerative diseases, our manuscript provides the first proof-of-concept data for the possibility of active (and possibly passive) immunotherapy of C9orf72 ALS and FTD. This study may rekindle the interest in active vaccination

for other neurodegenerative diseases after many failed clinical trials with passive immunotherapy. Taken together, we provide not only additional insights into poly-GA toxicity but also suggest a new therapeutic strategy.

Minor comments: many references are incomplete

We apologize for the mistakes. We checked and updated all references.

Regarding statistical analysis, why ANOVA and not non-parametric tests?

We rechecked normality for all datasets and found that ANOVA is the appropriate analysis for all but Fig. 2A/B (due to sharp drop of assay time at 3 s and wide shoulder of slow TG mice) and the new data in Fig. EV5C/D. Thus, we switched to box-plot presentation and performed pairwise Wilcoxon rank sum Test with Benjamini-Hochberg adjustment of p-values in the modified Fig 2A (compare Table S2 for p-values and new Fig. EV4 for longitudinal presentation). Our conclusions hold also using non-parametric testing.

Referee #3 (Remarks for Author):

GGGGCC repeat expansion is the most common genetic cause of both ALS and FTD. One major pathogenic mechanism is the production of dipeptide repeat (DPR) proteins through unusual RAN translation, as first reported by Dr. Edbauer and his colleagues a few years ago (Science, 2013). Indeed, numerous studies in cell cultures and animal models have demonstrated the toxicity of different DPR proteins, such as polyGA, polyGR, polyPR etc. These findings suggest that targeting DPRs may be a promising therapeutic approach.

In this study, Zhou and colleagues use an active immunization approach to generate anti-GA antibodies in mice expressing (GA)₁₄₉-GFP. Remarkably, they found that microglia activation in the spinal cord was largely prevented, although the reduction in polyGA inclusions was modest. This prove-of-concept study should be published (as a Correspondence, since it has limited scope), because it is the first report demonstrating beneficial effects of DPR-specific antibodies in the ALS/FTD field, although similar approaches have been used numerous times for other neurodegenerative diseases.

In this study, they used animals that overexpress polyGA. It is expected that targeting polyGA should have some beneficial effects since it is overexpressed polyGA that causes these ALS/FTD disease-like phenotypes. It would be nice if they could demonstrate a similar beneficial effect in C9ORF72 BAC transgenic mice, where different DPR proteins as well as repeat RNAs are co-expressed and it is unclear which molecule causes neurodegeneration, such as the one reported by Dr. Ranum that shows some strong disease phenotypes. However, it is not reasonable to demand the authors to perform this time-consuming experiment at this stage, which will significantly delay the publication of this prove-of-concept study. Maybe it is feasible if they could please show whether polyGA level in the CSF is reduced after immunization, if so, the result would further strengthen their conclusion that rescue of motor function and microglia activation is largely due to extracellular polyGA.

We thank this reviewer for the encouragement and enthusiasm. We now mention in the discussion that repeating these experiments in C9orf72 BAC models will be critical, but this experiment is unfortunately too time-consuming for a revision.

We also agree that measuring poly-GA reduction in the CSF would improve the translational impact of our study, but we so far failed to detect any poly-GA in the CSF of our mice (and patients), although our immunoassay clearly detects reduced poly-GA levels in brain homogenates. Thus, we cannot make direct conclusions on extracellular poly-GA levels. The dominant effects of Ova-(GA)₁₀ immunization on inflammatory changes as shown by transcriptomics and immunohistochemistry suggest that the induced antibodies reduce microglia activation presumably by acting on extracellular poly-GA or reducing neuronal damage by acting on intracellular poly-GA (see revised discussion).

Importantly, we provide new exciting data on neurofilament light chain (NFL) as clinically established biomarker for neuroaxonal damage. Importantly, GA-CFP mice show elevated NFL levels in CSF, which are reduced upon OVA-(GA)₁₀ immunization (new Fig. 3F).

1. *The first sentence in the abstract does not seem to belong there. The point is better made in the Introduction and Discussion.*

We removed the first sentence of the abstract as suggested.

2. *In the Introduction section, "neuronal cytoplasmic inclusions" should be "neuronal inclusions" since both nuclear and cytoplasmic inclusions have been reported.*

We changed the text as requested.

3. *In Figure 1B, could the authors please use different colors for mice with different treatment? Since they are all black and white, it is not obvious to tell TG-Ova-GA10 from others.*

We modified Fig. 1B and all other relevant figures (revised Fig 2A, EV1 and EV4) to include a consistent color code for the six different experimental groups.

4. *Figure 1C only shows a few images. This reviewer would like to suggest including some sort of quantification, which should not be difficult to do. For instance, they could count the number of neurons with inclusions per 100 cells in both control and patient brains. Also, it would be more convincing if additional brain tissues are examined, since the antisera give rise to high background signals, and/or a GA peptide blocking experiment is carried out to demonstrate the specificity of the Ova-GA10 antiserum.*

This is an excellent suggestion. We quantified the number of poly-GA positive neurons per 100 cells (new Fig. 1D). Additionally, we show colocalization of poly-GA aggregates labeled with a commercial rabbit anti-GA antibody and our mouse sera (new Fig. EV3). Finally, we performed blocking experiments using GST-(GA)₁₅ antigen to confirm specificity (new Fig. 1E).

5. *In Figure 2B, the statistical difference between PBS and GA15 should be presented in the panel as well.*

We included this important information in the revised Fig. 2A and Table S2.

6. *In Figure 3D, it is surprising to see 20% of cells in control mice show TDP-43 cytoplasmic localization. Is this number consistent with what the authors published before? In TG mice, the number is about 40%, which seems to be quite high. As Zhou and colleagues stated here: "poly-GA triggers modest TDP-43 phosphorylation and partial mislocalization of TDP-43". In contrast, the images in Figure 3C show a much lower % of cells with cytoplasmic TDP-43.*

We apologize for the confusion. We performed a blinded analysis of cytoplasmic TDP-43 mislocalization rather than of mature aggregates (which are not found in these mice). We now clearly marked the cells we counted for the analysis in the revised figure (now EV5E). In our initial characterization of the GA-CFP mice we had focused on phosphorylated TDP-43 using an immunoassay performed by collaborators (Schludi et al, 2017).

7. *On Page 5, what does "ADD significance!" mean? It seems to be an error.*

We removed this editing comment from the revised manuscript.

2nd Editorial Decision

11 November 2019

Thank you for the submission of your revised manuscript to EMBO Molecular Medicine. It's my pleasure to let you know that we have now received the enclosed reports from the referees that were asked to re-assess it. As you will see the reviewers are supportive of publication and we will be able to accept your manuscript pending final editorial amendments.

I look forward to reading a new revised version of your manuscript as soon as possible.

***** Reviewer's comments *****

Referee #1 (Remarks for Author):

The authors adequately addressed the concerns raised by this referee

Referee #2 (Comments on Novelty/Model System for Author):

The etiopathogenesis is not fully established but the choice of this model does not affect the quality of the work

Referee #2 (Remarks for Author):

The authors have answered most of the comments

2nd Revision - authors' response

29 November 2019

Authors made the requested editorial changes.

Corresponding Author Name: Dieter Edbauer

Journal Submitted to: Embo Mol Medicine

Manuscript Number: EMM-2019-10919-V2